# Fiber Bragg Gratings Sensors for Aircraft Wing Shape Measurement: Recent Applications and Technical Analysis

**DOI:** 10.3390/s19010055

**Published:** 2018-12-23

**Authors:** Zhen Ma, Xiyuan Chen

**Affiliations:** Key Laboratory of Micro-Inertial Instrument and Advanced Navigation Technology, Ministry of Education, School of Instrument Science and Engineering, Southeast University, Nanjing 210096, China; 18255183270@163.com

**Keywords:** aircraft, wing shape measurement, FBGs, accuracy

## Abstract

The safety monitoring and tracking of aircraft is becoming more and more important. Under aerodynamic loading, the aircraft wing will produce large bending and torsional deformation, which seriously affects the safety of aircraft. The variation of load on the aircraft wing directly affects the ground observation performance of the aircraft baseline. To compensate for baseline deformations caused by wing deformations, it is necessary to accurately obtain the deformation of the wing shape. The traditional aircraft wing shape measurement methods cannot meet the requirements of small size, light weight, low cost, anti-electromagnetic interference, and adapting to complex environment at the same time, the fiber optic sensing technology for aircraft wing shape measurement has been gradually proved to be a real time and online dynamic measurement method with many excellent characteristics. The principle technical characteristics and bonding technology of fiber Bragg grating sensors (FBGs) are reviewed in this paper. The advantages and disadvantages of other measurement methods are compared and analyzed and the application status of FBG sensing technology for aircraft wing shape measurement is emphatically analyzed. Finally, comprehensive suggestions for improving the accuracy of aircraft wing shape measurement based on FBG sensing technology is put forward.

## 1. Introduction

The USA is the birthplace of aircraft. In recent decades, the USA aircraft research and development technology has been leading the world and different aircraft used in various fields have been developed. The National Aeronautics and Space Administration (NASA) of the USA and aviation environment company jointly developed a solar powered UAV named “Helios”. As shown in Figure 1, its main function is to verify the feasibility of the long time flight of the solar powered aircraft. However, in 2003, the Helios suddenly broke up in the air during a test flight to validate fuel cell technology. During the flight test, due to the upward bending of the two wings caused by turbulence, the whole wing was pitched and oscillated heavily, which exceeded the distortion limit of the aircraft structure and finally disintegrated in the air [1,2,3]. The investigation results show that the main reason for the wing disintegration in the air is the use of inappropriate analysis methods to evaluate the wing layout design, which makes Helios very sensitive to turbulence. This case reminds us of the importance of structural design and real-time dynamic measurement of the aircraft during flight.

With the continuous development of aviation industry, the structure design of modern aircraft is developing towards the direction of large-scale and complex. Therefore, the requirements of aircraft design are more stringent, which makes the working environment of aircraft structure tend to be complicated, and the forms of deformation are becoming more and more diversified. In addition, the problems such as excessive deformation amplitude and low service life of wing caused by impact and vibration are becoming more and more prominent, which put forward higher requirements for deformation detection of aircraft structure.

As shown in Figure 2, to test aircraft wing shape sensing, the NASA Langley Research Center conducted first flight validation testing in 2008 [3]. The whole experiment conducted 18 flights tests and recorded 36 flight-hours. Researchers used fiber optic and strain gages in the test and performed multiple flight maneuvers. The results show that Fiber optic and conventional strain gages have excellent agreement and the FBG system performed well throughout entire flight.

As shown in Figure 3, under the background of fiber optic sensing technology, the USA Marine Corps has begun to use a new fatigue life tracking method to record the real flight loads on the aircraft, which can effectively track the structural dynamic deformation of the aircraft and help to extend the service time of the aircraft [4]. The USA Marine Corps hopes to track the deformation of the aircraft in real time without major structural changes after the full adoption of the new deformation measurement algorithm and then let AV-8B harrier jet able to serve until 2030.

Less structurally-rigid wings could be critical to future long-range, fuel-efficient airliners and designing aircraft with highly flexible, lightweight performance has good application prospects. As shown in Figure 4, the NASA Langley Research Center tested the performance of fiber optic sensing system (FOSS) on X56 in 2017 [3]. Researchers used 2D shape sensing to measure wing deflection and to make sure if the flutter modes exits and the results verified the excellent characteristics of FOSS in wing shape measurement.

Real time online detection of aircraft wing deformation plays an important role in ensuring future safe operation of aircraft. Researchers propose that the embedded intelligent sensors can be used to monitor the internal condition of aircraft in real time, which is very important for monitoring the dynamic deformation of aircraft wings and aircraft safety [5,6]. While the traditional online deformation measurement system brings convenience, the additional weight caused by strain gages, signal transmission cables and signal processing units cannot be ignored. Therefore, in the deformation detection and measurement process of aircraft, researchers have done a lot of research on the application of the new generation of FBGs in aircraft wing shape measurement. The cost performance of its reliability guarantee system has great advantages and development prospects.

FBGs has been widely used in deformation measurement of large structures due to its small size, light weight, strong anti-electromagnetic interference ability and self-tuning function [7,8]. FBG uses wavelength coding, which is slightly affected by the fluctuation of system light source and external factors [9]. It is suitable for online measurement of stress, strain and temperature outside or inside the structure in a complex environment. Moreover, with the development of science and technology, the manufacturing and application technology of carbon fiber materials is more and more mature, and it is widely used in aerospace structures because of its advantages of light weight, high strength [10,11,12,13]. Thus, researchers have done a lot of research on the application of FBGs in aerospace field.

Through the above-related introduction, this paper takes the FBGs as the research object. Firstly, the paper introduces the structure principle, sensing principle, FBG multiplexing technology, bonding process analysis and the layout optimization of sensors. Then, the research status of aircraft wing shape measurement technology based on different methods is analyzed. Finally, a summary is made to make recommendations for aircraft wing shape measurement and monitoring system based on FBGs.

## 2. Working Principle and Technical Characteristics

In 1978, K.O. et al. of the Canadian Information Research Center first discovered the photosensitive effect of germanium-doped fiber, and successfully developed the world’s first fiber grating by standing wave method [14]. In 1988, Meltz et al. of the East Hartford Joint Technical Research Center used two coherent ultraviolet beams to write hydrogen-loaded fiber into FBGs by side-exposing interference fringes. The fabrication technology makes Fiber Bragg grating have potential application value [15]. In 1993, Hill et al. put forward the phase mask method, which promoted the industrialization of Bragg grating applications, and promoted the application of Bragg grating in the sensing field [16].

### 2.1. Structure Principle of Fiber Optic

The optical fiber technology can be divided into single-mode fibers and multi-mode fibers. According to the different application scenarios, the single-mode fibers can be divided into helically wrapped single fibers, parallel fibers, orthogonal fibers and so on [13]. Multi-mode fibers are usually used for 3D shape sensing and Lally et al. [17] first used helical multi-mode fibers for real-time shape sensing with OFDR Rayleigh scattering interrogation technique. However, the FBGs are generally only possible in single-mode fiber. As shown in Figure 5, ordinary fiber optic structures generally include fiber core, cladding and coating, which are basically the same as the fiber materials used in communications. Among them, the fiber core plays the role of transmitting light signals. Its diameter is in the range of 5-50 μm. Its main composition is silicon dioxide, which is doped with germanium dioxide to improve its refractive index n_1_. The diameter of the cladding is about 125 μm of silica. The cladding needs to be doped with dopant to reduce its refractive index n_2_. The coating material is generally polymer composite material, which can realize the bending and corrosion resistance of the fiber core and its outer diameter is about 250 μm [18,19]. The schematic diagram of the fiber optic is shown below.

The process of transmitting light in fiber optic obeys the principle of total reflection. The refractive index n_1_ of the fiber doped with dopant is larger than that of the cladding refractive index n_2_. According to Fresnel’s law, when the incident angle of light incident on the fiber interface is greater than the critical angle, total reflection occurs at the interface between the fiber core and the cladding, and all the light propagates along the fiber core [20]. The schematic diagram of the propagation of light in fiber optic is shown in Figure 6.

Ordinary fiber optic sensors can only transmit signals. These sensors are non-intrinsic fiber optic sensors. FBGs cannot only transmit signals, but also sense signals. It is an intrinsic fiber optic sensor. FBG is written by ultraviolet rays in the fiber core, the written grating area has a strong sense of physical signals in the external environment. For reflective gratings, when the light source passes through the grating area, the wavelength is equal to the amount of light reflected back from the Bragg center wavelength, and the light of other wavelengths passes through the grating area [21]. The schematic diagram of the FBGs is shown in Figure 7.

### 2.2. Working Principle of FBG Sensor

The working principle of FBG sensor is actually to use the changes of the physical environment around the FBG sensor such as stress, strain and temperature to form the grating period or fiber core refractive index changes. It makes the central wavelength of Bragg grating shift, and a mathematical model is established through the center wavelength shift and measurement. Then the variation of stress and strain can be obtained.

As shown in Figure 8, when a broadband light source is incident to FBG, a small part of the incident light is reflected back at the corresponding wavelength due to the periodic refractive index of the FBG, and the rest of the light will be transmitted, thus the FBG acts as a light filter. According to the fiber coupled mode theory, the reflection wavelength of Fiber Bragg grating satisfies the following formula:(1)$$\lambda_{B}=2n_{eff}\Lambda$$
where *λ_B_* is Bragg wavelength, *n_eff_* is refractive index and Λ is Bragg period. 

The main factors affecting the center wavelength shift are temperature and strain, the formula [22] can be expressed as:(2)$$\frac{\Delta \lambda_{B}}{\lambda_{B}}=(1-P_{e})\varepsilon +(\alpha +\zeta )\Delta T$$
where Δ*λ_B_* is Bragg wavelength shift, *Pe* is the photo-elastic coefficient of the fiber core material, *ε* is the longitudinal strain, *α* is the thermal expansion coefficient of the fiber optic, *ξ is the* thermo-optic coefficient and *ΔT* is the temperature variation.

If the experiment is conducted at constant temperature, Equation (2) can be re-written as: (3)$$\frac{\Delta \lambda_{B}}{\lambda_{B}}=(1-P_{e})\varepsilon$$

It can be seen that the Bragg wavelength shift is linear with the strain without considering the effect of temperature.

### 2.3. FBG Multiplexing Technology

In 1996, M A Davis et al. [24] first applied FBGs wavelength division multiplexing technology to shape sensing and amplitude analysis. They mounted the fiber optic cable on the cantilever beam and detected the overall bending shape of the beam based on the strain detected in the fiber optic. In order to measure the deformation of large aircraft wing, distributed FBGs measurement method can be used to realize sensing monitoring. Distributed strain sensing based on Rayleigh scattering can be divided into optical time domain reflectometry (OTDR) and optical frequency domain reflectometry (OFDR). OTDR is the simplest way for strain measurement of interrogation technique. The principle of OTDR is based on launching a laser light pulse and collect Rayleigh backscattered light from the same fiber sensor. The principle of OFDR is when a continuous laser source is swept in frequency domain in period time, the collected backscattered light then generates data comparable to OTDR results. In the process of using distributed strain sensing measurement, researchers have found two kinds of fiber grating multiplexing technology, which can simultaneously measure the parameters of different measuring points. The technique that allows multiple FBGs with different Bragg wavelengths to be interrogated by different laser wavelengths is known as wavelength division multiplexing (WDM) and the technique that allows multiple FBGs connect with a light source is known as spatial division multiplexing (SDM) [25].

#### 2.3.1. Wavelength Division Multiplexing Technology

As shown in Figure 9, principle of wavelength division multiplexing (WDM) technology is that FBGs are connected in series through fiber optic, and each FBG sensor is arranged in the required monitoring position [26].

The advantage of WDM is to reduce the number of FBG channels, simplify the sensor network and save the layout space. The disadvantage of WDM is its weak anti-jamming capability. To avoid overlaps of FBG central wavelength to interrogate sensors, the number of FBGs it limited to 10–100 s in the same fiber when using WDM [25]. Moreover, if the fiber optic connected by FBGs is damaged, the whole demodulation channel will be damaged [27,28]. Sometimes it will affect the data measurement of other channels and reduce the accuracy of sensor measurement. Therefore, when using WDM technology to measure the structural deformation of aircraft wings, we need to pay attention to the fact that each FBG sensor needs to be unfolded naturally and cannot be bent. The offset of the center wavelength between adjacent FBGs on the fiber string must be less than the difference between the center wavelengths of two FBGs. In addition, the maximum offset of the center wavelength of the first FBG sensor cannot intersect with the minimum offset of the second FBG sensor center wavelength, which can satisfy most practical measurement applications.

#### 2.3.2. Spatial Division Multiplexing Technology

As shown in Figure 10, the principle of spatial division multiplexing (SDM) technology is to connect multiple FBGs in series on multiple channels to realize the measurement of different measurement points in the structure space [29,30,31].

The advantage of SDM is that the parallel topology is adopted and each channel is independent and non-interference. Even if one of the measurement channels is damaged, the whole measurement system will not be affected. This method eliminates the mutual influence between FBGs, reduces the risk of channel failure, and increases the measurement intensity. However, the disadvantage of SDM is that a large number of demodulation channels are occupied and the efficiency is reduced [32,33,34].

These two methods have their own advantages and disadvantages. In the experiment of measuring wing, according to the structure size and measurement requirements of the measured wing, we can combine two methods and arrange FBG sensor reasonably. Through the two methods complement each other, make up for each other’s shortcomings, so as to achieve better measurement results.

## 3. Influence Factors of FBGs Strain Transfer Efficiency

High precision measurement of stress and strain is very important for dynamic deformation measurement and evaluation of aircraft wing. The diameter of fiber core of is very thin and sensitive. In order to improve the strain measurement accuracy and stability of FBGs, we need to analyze the factors that affect the measurement accuracy of FBGs, in which the bonding process of FBG sensor directly affects the accuracy of FBG sensor in the experiment.

Through the analysis, we can know that the strain transfer rate of FBGs is directly affected by the adhesive bonding process. Therefore, it is necessary to analyze the main factors affecting the strain transfer rate in the adhesive bonding process, so as to improve the strain transfer rate.

### 3.1. FBGs Strain Transfer Coefficient

In the adhesive bonding process of FBGs, the grating area of FBGs is deformed synchronously with the adhesive bonding position due to the constraint effect of the bonding layer, and the strain rate of them is equal under ideal conditions.

As shown in Figure 11, the adhesive bonding mechanical transfer model [35,36,37,38] consists of FBGs, adhesive layer and measured structure, and the model is shown as follows:

Assuming that the fiber grating is only axially deformed and ignoring the Poisson effect, the formula [38] of strain transfer rate can be expressed as:(4)$$\alpha (x)=\frac{\varepsilon_{f}(x)}{\varepsilon_{m}}=1-\frac{\cos(hkx)}{\cos(hkL)}$$
where *α(x)* is the strain transfer rate, *L* is the half length of the grating area, *k* is the coefficient determined by the ratio of the elastic modulus of the fiber to the coating layer, *h* is the thickness of the adhesive layer, *ε_f_ (x)* is the strain of fiber core, *ε_m_* is the matrix strain at the adhesive site, *x* is a certain position in the axial direction of the grating area.

According to the above formula, for variable *x*, we can conclude that the strain transfer rate is inversely proportional to the distance to the gate area. The closer to the center of the grating, the greater the strain transfer rate is. When it comes to the center of the grating area, the strain transfer rate reaches the maximum. The average strain transfer rate over the entire length of the grating area is:(5)$$\bar{\alpha }=1-\frac{\sin(hkL)}{kL\cosh(kL)}$$

### 3.2. Influencing Factors of Strain Transfer Rate

The average strain transfer rate of the sensor is related to the length of the grating area and the thickness of the adhesive layer. In addition, it is also related to the elastic modulus of the adhesive layer and Poisson ratio, the elastic modulus of the coating and Poisson ratio.

Through experiments by researchers, the relationship between influencing factors [39,40,41,42,43] and strain rate is deduced and shown as follows in Table 1.

Through the proportional relationship in the above table, we know that the thickness of the adhesive layer is inversely proportional to the strain transfer rate of the sensor. That is to say, the smaller the thickness of the adhesive layer is, the higher the transmission rate will be. However, in practical application, the adhesive layer must have a certain thickness, otherwise, the adhesive layer will fall off easily. If the thickness of the adhesive layer is too large, the transfer rate will be too low to affect the accuracy of the sensor. Therefore, the allowable range of adhesive thickness to meet the working conditions is determined before operation. Then, on the basis of being firm, the adhesive thickness is as small as possible.

The strain transmission rate of FBG sensor is proportional to the grating length. In other words, the greater the grating area length is, the higher the strain transfer rate will be. Although the length of grating area becomes longer, the strain transfer rate will be higher, which will lead to the measurement area becomes longer, and the strain of a certain point cannot be accurately measured. It will make the measurement accuracy lower. On the one hand, the long FBG sensor is damaged easily by mechanical collision during operation, which makes the sensor layout and measurement inconvenient. On the other hand, if the length of grating area is too long, it will lead to uneven coating and uneven thickness of the adhesive layer. Therefore, it is necessary to increase the grating area length of the sensor to increase the strain transfer rate on the basis of selecting the appropriate length of adhesive layer and avoiding the excessive length of the measuring area.

## 4. Comparative Analysis of Other Measurement Methods

The method of aircraft wing shape measurement has been improving and diversifying along with the development of aircraft. In 1922, A. Baumhaluer and C. Konin proposed a mass balance method that effectively prevents the control surface flutter. In the late 1920s, H.G. Kussner, W.J. Duncan and R.A. Frazer summarized the wing flutter problem and established the corresponding theoretical basis. Then T. Theodorson proposed a feasible method to solve the flutter problem on the basis of obtaining the exact solution of the resonant aerodynamics in the wing control surface in 1934. In the 1950s, supersonic aircraft began to develop continuously and researchers focused on the deformation measurement and flutter theory of delta wing and small aspect ratio swept-back wing. By the year 70s, the emergence of computers brought great changes to the traditional aeroelastic theory [44]. The test method of flutter airworthiness has also undergone profound changes, which represents the arrival of subcritical test technology.

This section first introduces and analyzes the working principle, advantages and disadvantages of strain gages measurement, laser measurement and visual measurement in aircraft wing shape measurement. Then, we summarize the advantages and disadvantages of these methods, highlighting the advantages of FBGs in measurement.

### 4.1. Measuring Deformation with Strain Gages

For real time load measurement related to aircraft wings, the commonly used method is a relatively extensive "strain gages/load calibration" method developed in the 1950s. 

As shown in Figure 12, a series of discrete traditional strain gages are installed on the wing, and a series of concentrated loads are applied on the wing surface to calibrate the response of the strain gages. Then the measured values are input into the software to run the results of the load formula in real time. This method is time-consuming, energy-consuming, cost high and large volume [45,46,47].

### 4.2. Measuring Deformation with Laser

The principle of laser measurement is to use laser beam as a signal source to illuminate the reflection target of the wing surface. Then, the reflected beam is transmitted to the signal processing system through an optical receiving system and a photodetector. The system measures the target point by point and obtains the deformation parameters of the wing surface according to the distance or other information [48,49].

As shown in Figure 13, the measurement method [50] is to perceive the deformation of the wing structure by application of laser scanning and Image Pattern Correlation Technique (IPCT) [51,52]. This method has the advantages of high accuracy and good real-time performance. However, the measuring method is based on laser source, and the life of laser source is shorter in long-term use. It is difficult to realize the full field deformation measurement of large structures.

### 4.3. Visual Deformation Measurement 

With the development of computer vision, the visual measurement method for aircraft wing deformation is proposed and applied step by step. The visual measurement method has many advantages, so it has been widely studied by researchers. It can be divided into several categories according to the difference of the measured patterns.

The NASA Langley Research Center developed a video model deformation (VMD) technology for high speed wind tests [53,54]. VMD technology is a digital photogrammetry technology which uses CCD camera to record and process digital video images. The technique has been a research hotspot for the last 30 years and has been applied to deformation measurement of wind tunnel models. As shown in Figure 14, it fixes the LED feature points on the F/A-18 wing model and uses Optotrak Flight Flexibility Measurement System to measure these points. The deflection curves of the wing are measured by the system. This method is not only easy to implement, but also has high accuracy. However, this method still has the shortcomings of insufficient measurement points and cannot make full use of resources in flight test, so it is often used in wind tunnel test or some ground static test [55,56,57,58].

In order to obtain the deformation information of the full field, Fleming, G. A. et al. [59] carried out the deformation measurement of the model by Projection Moiré Interferometry (PMI) on the wing surface in 1999. As shown in Figure 15, this method is based on the VMD measurement system. On this basis, the measurement range is extended, and the full field distribution of the wing surface deformation is obtained. Through the system measurement scheme and measurement results, it can be seen that the method overcomes the problem of few measurement points, and can obtain the distribution of the full field deformation. However, it is susceptible to the influence of illumination changes, and the measurement environment is limited, so it is only suitable for some specific ground deformation measurement [59,60,61,62,63,64].

In recent years, the wing shape measurement method based on Image Pattern Correlation Technique (IPCT) [52,53] has been continuously studied and developed. This method not only has the advantages of full-field measurement, but also has the characteristics of non-contact and real-time. It can be used to measure the deformation of wing in flight theoretically. In order to put the method into production as soon as possible, the method was studied by the German Space Center (DLR) and other well-known power research institutes and a series of aircraft online testing technologies have been carried out. In 2013, a successful AIM^2^ flight test [65] was performed with the NLR Fairchild Metro II. Aim for the test flight was to test IPCT (shown in Figure 16). The method of the test has the advantages of flexible meshing and convenient acquisition of displacement vector field or strain field information. However, the measurement accuracy will be affected by noise and complex weather conditions when using high-speed camera. At the same time, the measurement accuracy and adaptability of this method need to be further verified due to the complexity of aircraft wing deformation in dynamic environment [66,67,68,69,70,71].

As shown in Table 2, we can know that the strain gages measurement method has the disadvantages of large volume, large additional load and high cost. The method of laser measurement has many shortcomings, such as complex device, poor stability, and cannot be measured in the full field. Although the visual deformation measurement has many advantages, it is still affected by site constraints and environment, which affects the accuracy of aircraft wing shape measurement. Therefore, it is of great significance to find a small volume, light weight, high precision, strong anti-jamming and high adaptability method for dynamic aircraft wing shape measurement. Many advantages of FBGs can meet the requirements of high-precision dynamic deformation measurement in complex environment. Therefore, researchers have done a lot of research on aircraft wing shape measurement based on FBGs in recent years and we will analyze it in Section 5.

## 5. Research Status of Aircraft Wing Shape Technology by FBGs

Researchers have done lots of research and experiments on FBG sensing technology in many fields of aircraft, especially in Structural Health Monitoring (SHM) field. For example, Ryu et al. [72] used strain monitoring method in wing structures by FBGs to monitor the buckling behaviour of a wing box in real-time in 2008. Kosters et al. [73] used surface-mounted FBGs to detect the impact damage for composite primary aircraft structures in 2010. Mieloszyk et al. [74] used FBGs in smart composites, they developed SHM systems to monitor shape and defects in composite skin of the aircraft wing in 2011. Gherlone et al. [75] used the strain-based deformation shape reconstruction by FBGs for SHM of aircraft structures in 2012. Minakuchi et al. [76] used embedded FBGs and life cycle monitoring system to detect the debonding in composite patches and lap joints in 2013. In the above examples, we can see that researchers have made a systematic study on the application of FBG sensing technology in SHM. Therefore, we can deduce that the application of similar techniques and methods to the research of measurement methods of aircraft wing shape sensing. In this section, several typical cases will be synthetically analyzed in order to provide ideas for the research of wing shape measurement.

In 2003, in order to verify the effectiveness of FBGs for aircraft wing health monitoring system, Jung-Ryul Lee et al. [77] of Korea Advanced Institute of Science and Technology (KAIST) designed a FBG system to test the dynamic strain of the 1/25th scale Boeing 737C wing model (shown in Figure 17) in the wind tunnel, and measured the the dynamic strain inside the wing. The FBG system is based on the high-power wavelength-swept fiber laser (WSFL) and signal processing unit. In the configuration of FBG system, to reduce the errors, a triangular waveform was used to modulate the FP filter and FPWL was used to eliminate the wavelength non-linearity of the WSFL output. In the process of electronic signal processing scheme, analog signal processing circuit was used to convert optical signal to electrical wavelength-encoded signal in the time domain and 20 MHz counter was used for the system. Besides these, all bare FBGs were re-coated by means of acrylate in which the birefringence phenomenon of each single mode fiber can happen easily. During the impulse test, a PZT sensor was chosen and bonded to compare natural frequencies with the results of other sensors and an electric strain gauge (ESG) was used as a dynamic strain sensor to verify the FBG. Researchers located reference FBG after the coupler because the use of the coupler induced 50% loss in the intensity of FBGs.

The results show that the dynamic strains measured by FBGs have a good agreement with ESG and PZT sensor [78] inside the wing and the power spectral density obtained by FBG shows good agreement with those as well. The dynamic strain test in the wind tunnel with various situations successfully the effectiveness of FBGs in the monitoring system. Through the analysis of the test, we can find that the researchers used several groups of tests to verify the effectiveness of FBGs. To avoid the interference of errors, researchers used WSFL, FPWL, analog signal processing circuit and other methods in the wind tunnel test. Noted that the flutter was detected by embedded FBGs which is a hazardous phenomenon for the aircraft wing structure. Therefore, in the research of wing shape measurement, the influence of flutter status on the accuracy of wing shape measurement and baseline measurement should be considered.

In 2005, NASA Langley Research Center published a report on aircraft wing shape measurements to mitigate accidents caused by system and component failure [79]. In this report, using a distributed fiber optic strain measurement system combined with beam theory [80,81,82] and inverse Finite Element Method (iFEM) [83,84] method for wing structural deformation. As shown in Figure 18, a new instrumentation based on principles of OFDR was developed for strain measurement with Fiber Optic Strain System (FOSS). The FOSS has the characteristic of large-scale, densely distributed Bragg gratings strain sensors. A FOSS fiber and 13 foil gauges were used to build an aluminum beam test article and the FOSS strain values was used for iFEM model to estimate deflection.

Researchers have carried out detailed design and analysis on the selection of experimental equipment and the layout of equipment. An aluminum bar with the material of 7076AL which measures 1219.2 mm × 63 mm × 6 mm was used as instrumented test article to validate the capability of FOSS and 333 Fiber Optics Bragg Grating sensors were used for measuring axial strain along the length of the instrumented test article. A thermal-based technique was used to determine first 4 individual grating locations and the rest locations were calculated through linear regression. In the bonding design of fiber optic, 13 single axis foil gauges were bonded along the centerline of the bar and foil gauges data acquisition system was calibrated at room temperature. In the fixed form of the instrumented test article, the mount screws were locked in place by the collar-screw located near the top of each post instead of screwed into the posts in which the fixed form has more flexibility. In the strain validity verification, the advantages and disadvantages of FEM and iFEM are analyzed and the experimental results show that iFEM has the advantages of simple mesh division, fast calculation speed, less than 5% deflection error and high measurement accuracy. Through the study of NASA Langley Research Center reports, we can see that the FOSS-iFEM technology has many advantages and potential application prospects. In this study, iFEM technology is a considerable application in wing shape measurement combining with FBG technology. This idea may help to improve the measurement accuracy based on FBGs, simplify the calculation process, and reduce the impact of additional loads. It may also suitable for measurement in a dynamic environment.

In 2006, Andrea Cusano et al. [85] of the University of Sanio proposed an aircraft wing model modal analysis experiment based on embedded FBGs. In this study, considering the effect of vibration on composite wing modal analysis, researchers used an instrumented impact hammer with the model of PCB 086C03 to provide mechanical impulsive excitation. The sensitivity of the hammer is 2.3 mV/N and the frequency is 8 KHz. Four piezoelectric accelerometers (ICP Mod. 35C22) were bonded to the bottom surface of the wing as reference sensing elements. The voltage sensitivity of the piezoelectric accelerometers is 10 mV/g, the broadband resolution is 0.0032 g·rms. Four FBG sensors were used as optical sensing elements. Researchers exploited a sensors interrogation system and a chirped and strongly apodized fiber grating was used which can reflect linearly varying with optical wavelength. Passive ratiometric technique was used to provide sensor output and all three signals have been synchronously detected and stored due to the passive nature of the proposed technique.

As shown in Figure 19, for systematic analysis of the experiment, a grid of 29 excitation white points has been selected along the composite wing for mode retrieving. The location of grid in this study is the better place for impact excitation than other places. Researchers took five times hammer impacts and acquired five excitation-output pairs signals in order to obtained more accurate data. FRFs and SFRFs determination took the same procedure. The analysis show that bonded accelerometers data and embedded FBGs can get similar shapes, which means that strain obtained by embedded FBGs can respond to changes in aircraft wing shape. However, in the process of SHM, this method is limited to bending modal analysis of simple beams and torsional modal analysis, it may limit the modal analysis of aircraft wings in complex environments, so Andrea Cusano et al. think it is necessary to develop a damage monitoring system for complex rhythmic structures based on active modal analysis which may also be beneficial for wing shape analysis.

In 2012, Hideaki Murayama et al. [86] of Japan launched a large-scale wing strain measurement experiment based on FBGs network. As shown in Figure 20, researchers used carbon fiber reinforced plastics as the material of composite wing box and the length is 6 m. Seven FBG arrays were equipped on the wing skin and each array had about 30 or 40 FBGs with 10 mm gauge length. In order to get the overall deformation of the wing box, sensing system based on OFDR was chosen and bonding eight long-length FBGs with 300 mm and six long-length FBGs with 500 mm around stress concentration to get more details on wing structural deformation.

As shown in Figure 21, in the layout of FBG arrays, the researchers equipped the wing box with seven FBG arrays (A-1–A-7), eight long-length FBGs with 300 mm gauge length (B-1–B-8), six long-length FBGs with 500 mm gauge length (B-9–B-14). Each array had about 30 or 40 FBGs with 10 mm gauge length and FBGs were arrayed in a 100 mm pitch along the fiber.

The stress distribution of tensile strain A-1 is obtained by combining finite element analysis with variable load. As shown in Figure 22, it can be seen that the measured and calculated values have a good match. The experimental results also show the stress distribution of B-1. It can be seen from the diagram that the stiffness of B-1 varies locally on the tip of the truss and the strain varies greatly. The case shows that FBG arrays have good performance in strain and load distribution monitoring of wing box structure and we can also estimate the applied load with high-precision by using strain data measured by FBG arrays. The major feature of this case is that FBGs are designed in several FBG arrays forms according to the needs on different parts of the composite wing box and this measurement scheme may also have a good effect on the measurement of wing shape.

In 2012, in order to accurately calculate the deformations of the aircraft wing, Yi Jincong et al. [87] of Shanghai University proposed an implementation method of FBGs in the deformation detection of wing structure. Researchers used epoxy resin plate with a trapezoid form as a kind of aircraft wing model. The dimensions of the wing model is 225 mm × 160 mm × 380 mm × 195 mm as shown in Figure 23a and the thickness is 1 mm. The experimental setup consists of FBG demodulator, computer, packaged FBG sensor array and three-dimension reconstruction visualization system as shown in Figure 23b. By comparison with optical other algorithms [88,89], an algorithm based on FBGs for deformation reconstruction of an aircraft wing model was proposed by researchers. The algorithm uses the curvature information measured by FBGs to reconstruct the wing shape. Firstly, the space coordinate system of the experimental model is established, and the coordinate values of each point in each curve are calculated according to the curvature. Then the smooth interpolation method is used to fit and reconstruct multiple curves. Finally, the shape of the surface can be reconstructed accordingly. By this method, the steady-state deformation and dynamic vibration characteristics of real-time deformed structures are experimentally studied. Researchers gave various static deformation in manual way for the wing model and the experimental results show that the reconstructed shape based on the designed algorithm have a good realistic rendering reality and real time effect. The characteristic of in this case is that researchers used the simplified wing model with few FBG arrays and OpenGL technology achieved the deformation reconstruction of the wing model. In the research of wing shape reconstruction, appropriate models and algorithms are helpful to improve the accuracy of wing deformation reconstruction. In this case, researchers used an algorithm based on curvature information combined with FBG arrays to reconstruct the wing shape in which is worth learning.

In 2016, Matthew J. Nicolas et al. [90] of the USA pointed out that most reports of aircraft wing shape measurement by FBGs were laboratory-level testing. As shown in Figure 24a, the researchers took a 5.5 m carbon composite wing of an ultralight aircraft as an example, designed a practical method to calculate the deflection and out-of-plane load of the wing shape by using a high density FBGs on a three-tier whiffletree mechanism [91,92]. In the layout of FBGs, two optical fibers along the main spar was instrumented, one on the upper skin and one on the lower skin. The sensor spacing of FBGs approximately 12.5 mm, 388 sensors on the upper surface and 390 sensors on the lower surface of the composite wing in total. Researchers set FBGs and strain gages (approximately 12.7 mm from the fiber) on the left wing and both left and right wings were set with a tip displacement gage. Labview software was used to obtain strains and applied loads and two channels was used to get data from 778 FBGs. In the algorithmic design of the composite wing, the deflection and load algorithms were used based on classical beam theory and considered only pure bending. Then, the deflected wing shape and the out-of-plane load accurately obtained by high spatial resolution FBGs measurement.

As shown in Figure 24b, the results of the experiment show that total computed load is within 1.62% of the applied load, the computed out-of-plane loads is within 2% of the applied load, the calculated deflections for several load cases, and the predictions are within 4.2% of the measured data, the computed FBG-based loads are within 4.2% of the measured applied loads. Through the experimental process and data results, we can see that high spatial density of the strains from FBGs can reveal some physical details not normally obtained from conventional strain gages. Compared with conventional strain gages, the method designed by the researchers with FOSS system [93,94] can provides a higher density (every 12.5 mm) of strain measurements and it can be used to determine the health of the wing by obtaining deformation data.

In 2017, Cui Peng et al. [95] of China proposed a feasible method to measure the deformation of aircraft wings based on FBGs as shown in Figure 25a. Compared with the conventional indirect measurement method, it uses a specific derivation process to measure the vertical displacement variation of the wing model. Researchers did experiments and simulations, and bonded FBGs in the locations of the wing with large strain variation to improve the accuracy of the experimental results. 28 FBGs were divided equally into 4 arrays and the central wavelength in each group were separated by at least 2 nm. Three arrays were arranged in the upper surface of wing model and one array in lower surface. The type of demodulator they chose is SM130.

In the constant temperature experiment, the change of wavelength is proportional to the strain. As shown in Figure 25b, Cui Peng et al. increased the load at point 0 gradually from 100 g to 1000 g with a gradient of 100 g and the test was repeated three times. Then the finite element model of the wing was established to simulate the relationship between the strain and the displacement. In the experiment, when the force is applied at point 0, the vertical displacements measured by the FBGs in experiment under different load pressures is lower than 5% of the average error compared with those directly measured and the liner fitting function of the experimental vertical displacement is close to the liner fitting function of the actual vertical displacement as shown in Figure 26. The study, in this case, combines finite element method with FBGs to verify the feasibility of FBG technology for measuring vertical displacement and this is a new attempt in the research of wing deformation measurement method.

In 2018, Hong Li et al. [96] proposed a new deflection monitoring method for wing spar I-beam using multi-elementoptical fiber grating rosettes and the method combined with Timoshenko beam theory and iFEM [84,85], the wing frame deformation was measured in real time. The tested wing used in the experiment is 1/4-scale model of a kind of aircraft and aluminum-magnesium alloy materials were used for spar cap and shear web. As shown in Figure 27, the FBGs were bonded with epoxy resin adhesive 3M DP. The maximum strain value of the wing spar during loading test is 1500με and the initial wavelength interval was selected greater than 2 nm. In the procedure of loading, the load is increased by 10 kg increments up to 100 kg and recorded the magnitudes of all FBG sensors and foil strain gages. In order to ensure the security, the load in some point is limited to 80 kg and loading of each point step three times and taking the average. Through the experiment, researchers concluded that theoretical and simulation results of the resistance gage value are consistent and the deflections of the wind spar were well estimated from the bonded FBGs. The study, in this case, pointed out that the FBG rosette can sufficiently monitor the wing structural condition even though they were positioned in the bonding points between the shear web and spar cap. The main feature of this case is that researchers used FBG rosette to measuring deflection, and the experimental results verify its feasibility. Deflection is an important measurement index in the measurement of wing shape and the study by Hong Li et al. Presents useful information to wing shape measurement system.

## 6. Conclusions

In this paper, the research progress and application of FBGs in aircraft wing shape measurement are reviewed. Combined with the basic principle and performance characteristics of fiber optic sensor, the multiplexing technology of FBGs is introduced. Besides, the paper introduces the adhesive bonding principle of distributed FBGs, analyzes the influencing factors of strain transfer rate in adhesive bonding process and suggestions for improving strain transfer rate are put forward. Next, the characteristics and limitations of existing methods for measuring aircraft wing deformation, such as strain gages measurement, laser measurement and visual measurement are introduced. Finally, eight examples of aircraft wing SHM and deformation measurement based on FBGs are selected for detailed analysis. According to different needs, researchers used different methods to apply to different aspects of the wing. Jung-Ryul Lee et al. verified the feasibility of FBGs and emphasized the influence of flutter status cannot be ignored. The study of NASA Langley Research Center reports showed the advantages and potential application prospects of FOSS-iFEM technology on the wing shape monitoring. The research of Andrea Cusano et al. Showed that active modal analysis is necessary to SHM of complex rhythmic wing structures and this idea can also be used to extract abnormal points in wing shape measurement. Hideaki Murayama et al. used specific FBG array layout and verified its well performance in wing box structure. The enlightenment of this example is that in order to give full play to the effect of FBGs in the shape measurement of wing box, we can design different FBG arrays layout according to the strain situation of different positions, so as to achieve measurement accuracy as high as possible. Yi Jincong et al. designed an algorithm based on curvature information and reconstruct the wing shape by FBG arrays. It shows that suitable reconstruction algorithm of the wing shape is necessary for the wing shape measurement. The research results of Matthew J. Nicolas et al. proved the superiority of FOSS system on aircraft wing shape measurement and high spatial density of the strains from FBGs can benefit to full-field deformation of wing shape measurement Cui Peng et al. measured the vertical displacement variation of the wing model by FBGs and FEM in the constant temperature condition. The experimental results prove the feasibility of the method. The characteristic of Hong Li et al. research results is that they used FBG rosette to measuring deflection of aircraft wing and proved its beneficial effect. It may be a good prospect in the field of wing shape measurement. Moreover, we can find that different types of adhesives are selected according to different experimental conditions in these cases and the representativeness of the data is improved by taking the average value to ensure the validity of the experimental data.

Although the literature referenced in this paper is limited and cannot cover all the fields of wing shape measurement. In view of these representative research results, this paper puts forward the following comprehensive ideas. For specific wing dimension, we need to determine the measurement methods we used. It is a good idea to apply FBGs and strain rosette to the full-field deformation measurement of wing shape. In order to improve the accuracy of the measurement system, the combination of FOSS system and iFEM can be adopted. In the details of the specific experimental procedures, the layout of sensors needs to be designed reasonably, which helps to achieve full-field measurement. Suitable material of adhesive is necessary for the FBG adhesive bonding and then determine the thickness of the adhesive layer and correlation coefficient, strictly adhere to the adhesive bonding process, and design the calibration method of FBGs. Good layout design, adhesive bonding process, and calibration method are helpful to improve the accuracy and stability of the measurement experiment. Combined with iFEM analysis, modal analysis and related algorithms, we need to study how the computer can reprogram the sensor network or accurately measure the wing dynamics when the FBG sensor at a certain position fails or damaged during the extreme flight conditions of an aircraft. Then it can be applied to transfer alignment and dynamic measurement of aircraft baseline to improve the measurement accuracy in a dynamic measurement environment.

## Figures and Tables

**Figure 1 sensors-19-00055-f001:**
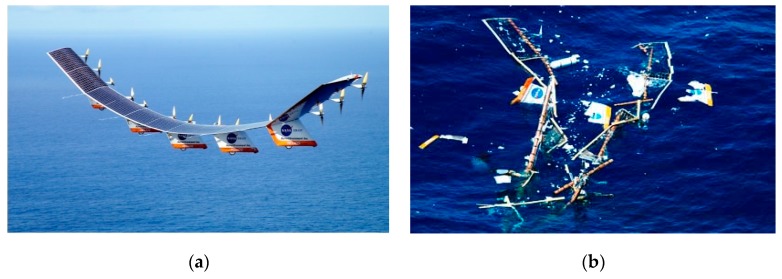
(**a**) Test photos of Helios Solar powered UAV; (**b**) Aerial disintegration photos of Helios [1,2,3].

**Figure 2 sensors-19-00055-f002:**
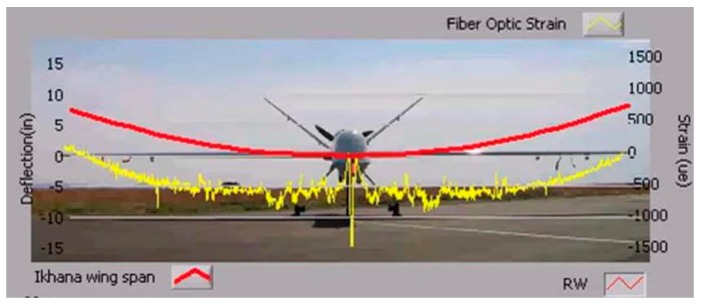
Real-time aircraft wing shape measurement using fiber optics sensors [3].

**Figure 3 sensors-19-00055-f003:**
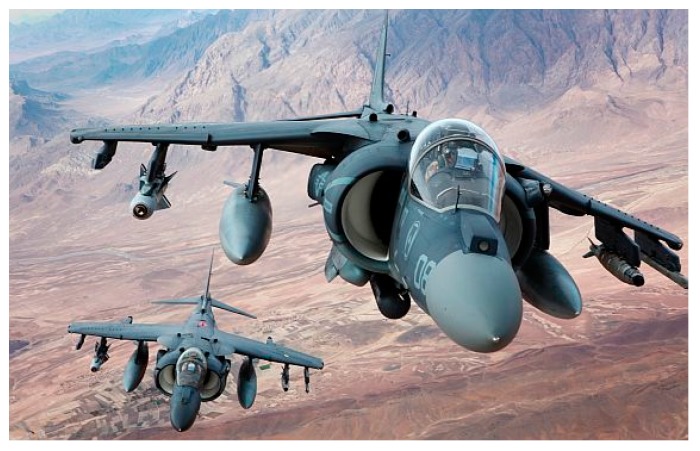
Fiber optic sensing technology applied to fatigue life tracking of AV-8B harrier jet [4].

**Figure 4 sensors-19-00055-f004:**
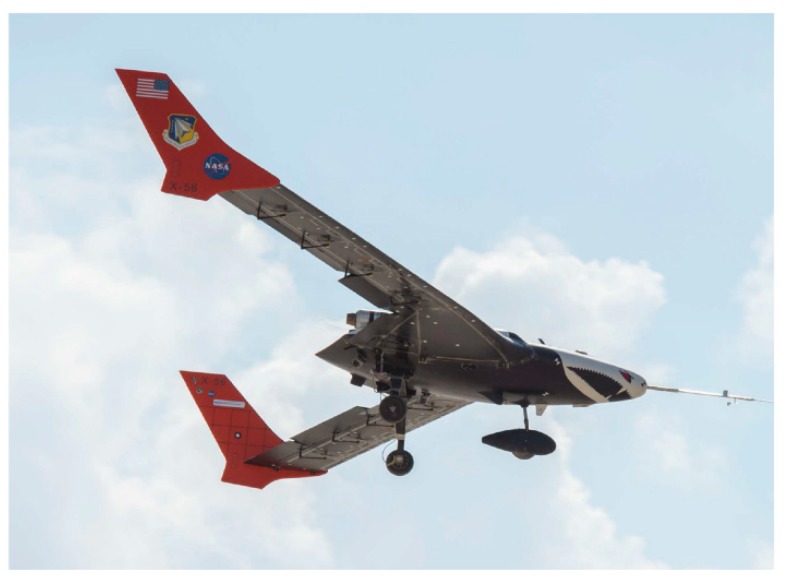
Performance test of fiber optic sensing system (FOSS) on X56 [3].

**Figure 5 sensors-19-00055-f005:**
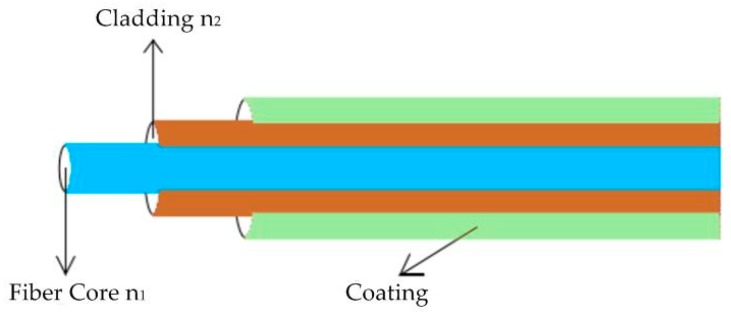
Schematic diagram of fiber optic.

**Figure 6 sensors-19-00055-f006:**
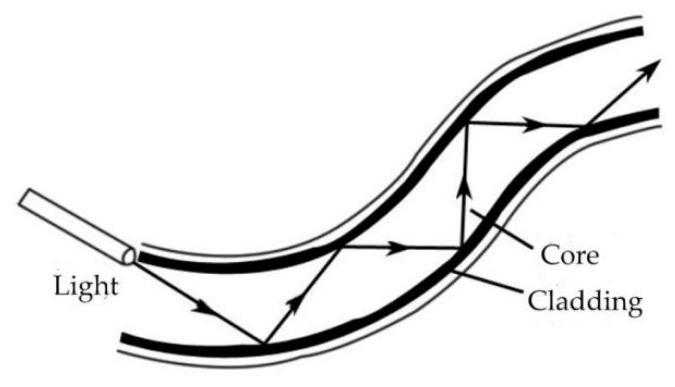
Schematic diagram of light propagation in fiber optic.

**Figure 7 sensors-19-00055-f007:**
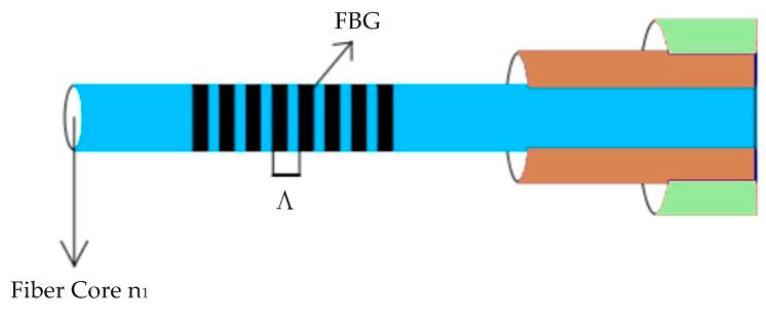
Schematic diagram of fiber Bragg grating (FBG) sensor structure.

**Figure 8 sensors-19-00055-f008:**
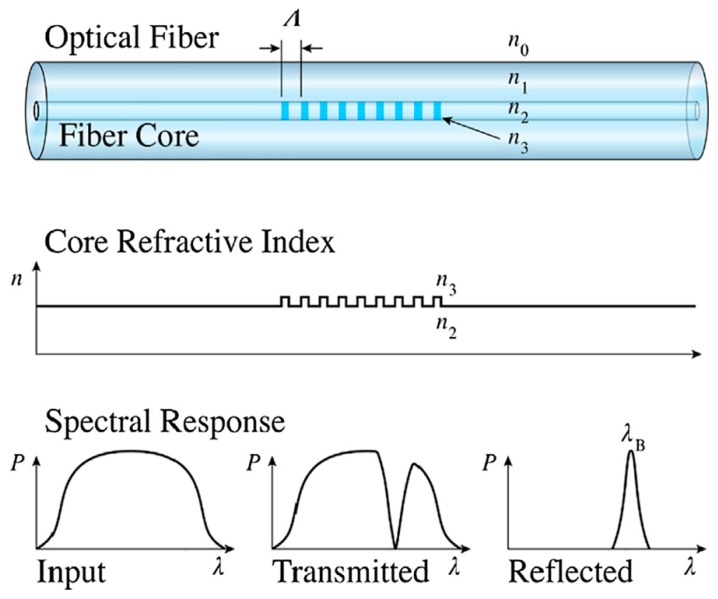
Working principle of FBGs [23].

**Figure 9 sensors-19-00055-f009:**
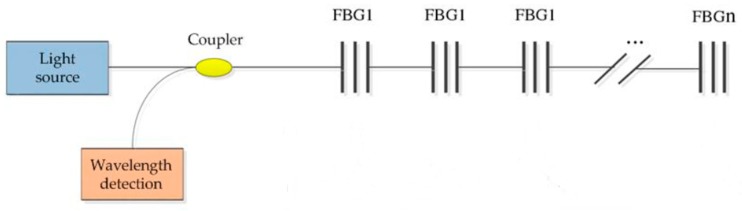
Schematic diagram of wavelength division multiplexing technology.

**Figure 10 sensors-19-00055-f010:**
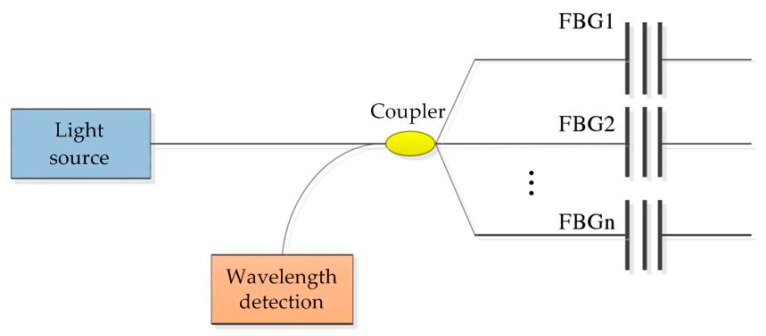
Schematic diagram of spatial division multiplexing technology.

**Figure 11 sensors-19-00055-f011:**
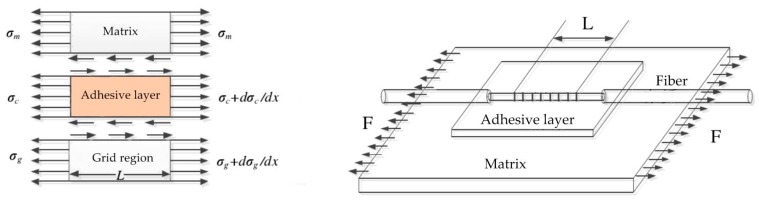
Mechanical transfer model of FBG sensor [38].

**Figure 12 sensors-19-00055-f012:**
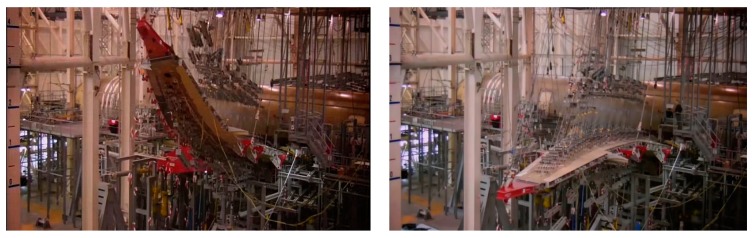
Detection of structural deformation through strain gages.

**Figure 13 sensors-19-00055-f013:**
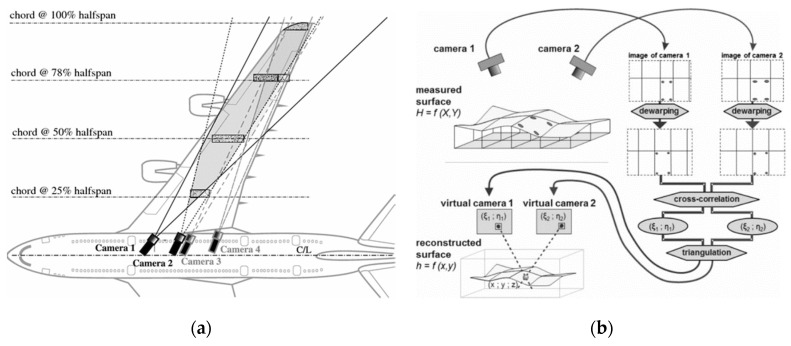
(**a**) Application of Image Pattern Correlation Technique (IPCT) cameras sets placed in aircraft for wing deformation measurements; (**b**) IPCT processing flow [50].

**Figure 14 sensors-19-00055-f014:**
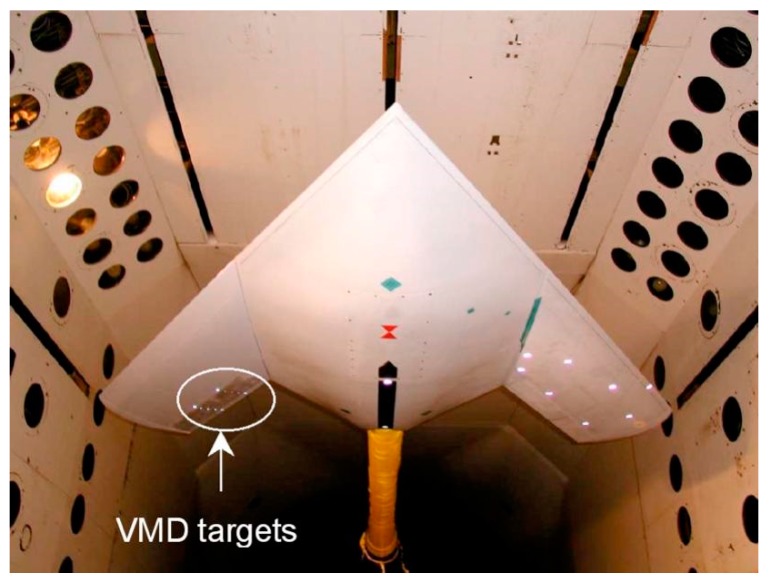
Video model deformation (VMD) measurement system [54].

**Figure 15 sensors-19-00055-f015:**
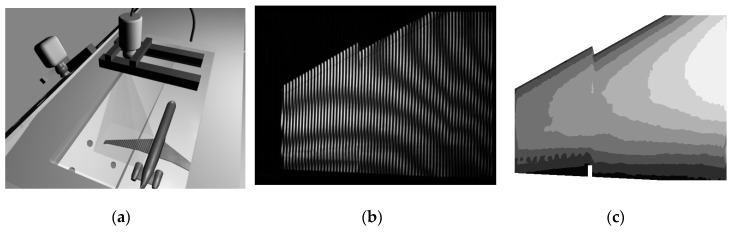
(**a**) measurement scheme; (**b**) Projection of mohr interference fringes; (**c**) Distribution of aircraft wing deformation [60].

**Figure 16 sensors-19-00055-f016:**
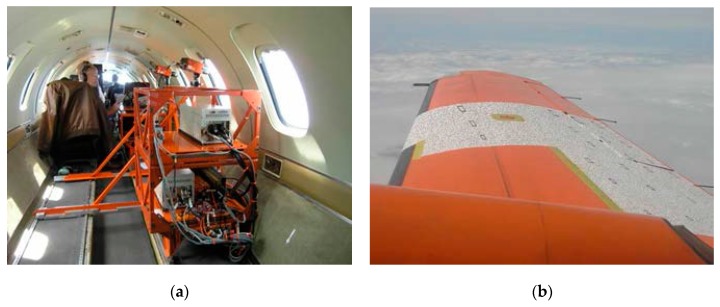
(**a**) The interior of the aircraft; (**b**) The partly speckled on the wing [65].

**Figure 17 sensors-19-00055-f017:**
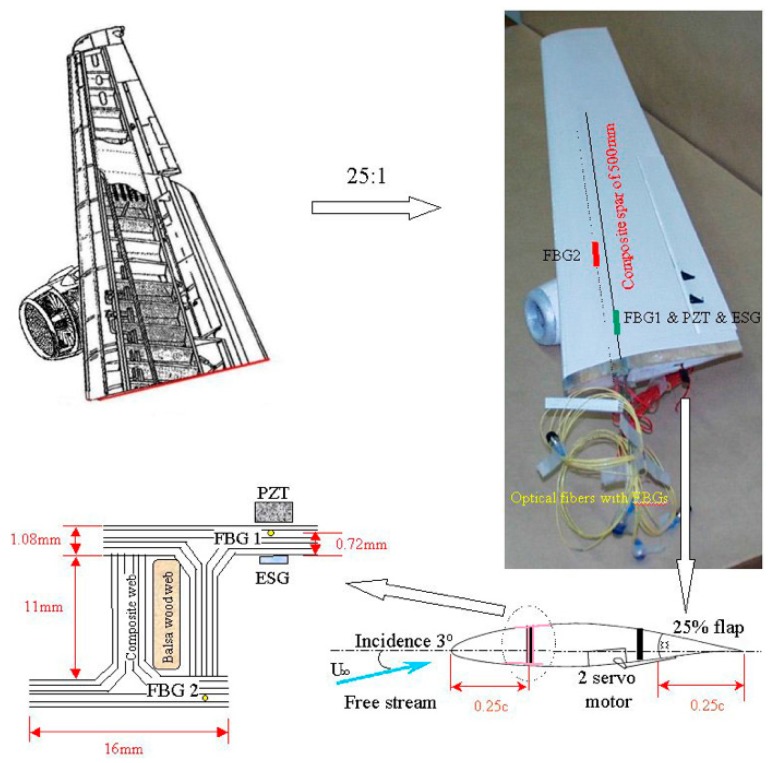
Configuration and geometry of the wing [77].

**Figure 18 sensors-19-00055-f018:**
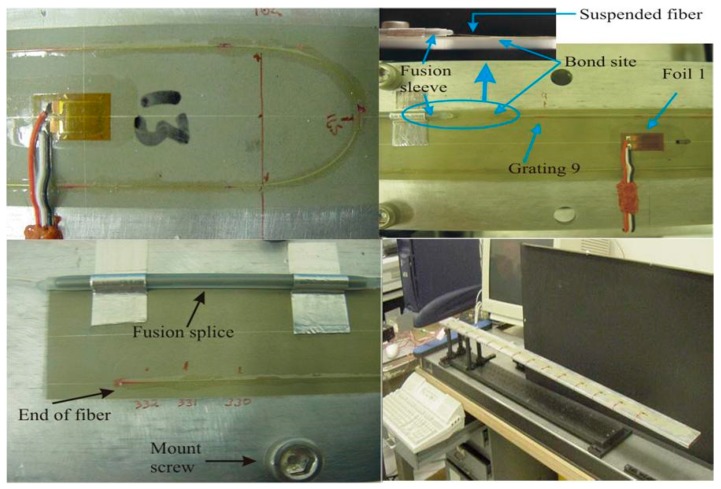
Experimental equipment and FOSS System Layout Design [79].

**Figure 19 sensors-19-00055-f019:**
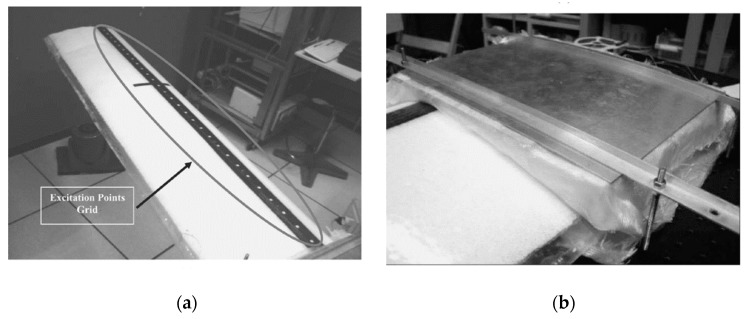
Two photographs of the wing. (White color) Body and (black) spar in (**a**); (**b**) a detail of the clamped end of the wing [85].

**Figure 20 sensors-19-00055-f020:**
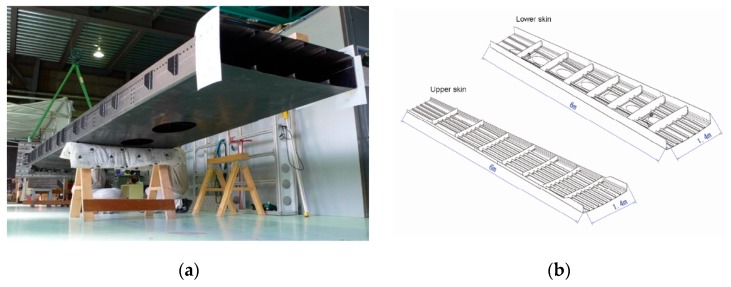
(**a**) Physical map of composite wing box; (**b**) Internal structure of wing box [86].

**Figure 21 sensors-19-00055-f021:**
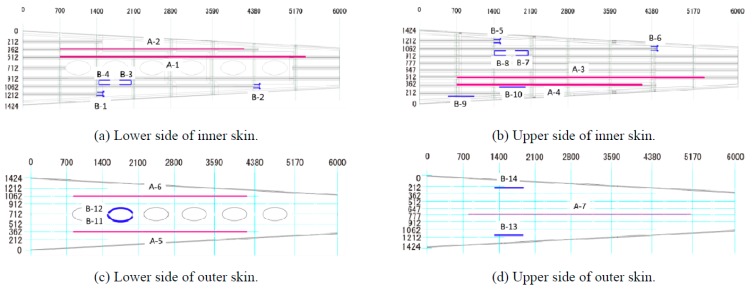
Array layout of FBGs [86].

**Figure 22 sensors-19-00055-f022:**
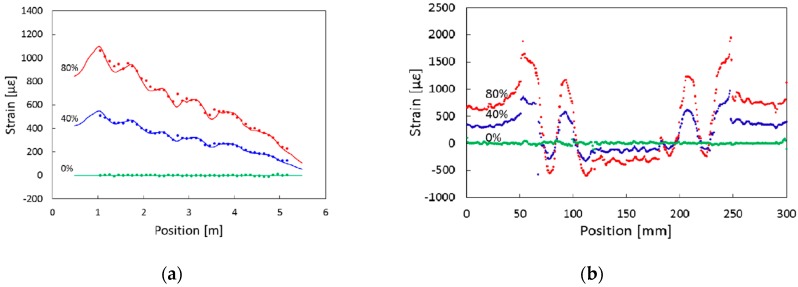
(**a**) Strain distributions measured by A-1; (**b**) Strain distributions measured by B-1 [86].

**Figure 23 sensors-19-00055-f023:**
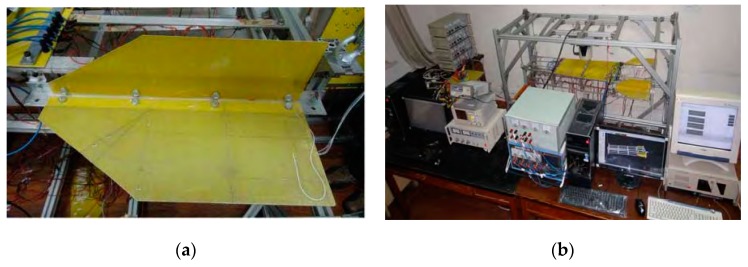
(**a**) Experimental model; (**b**) The whole experimental setup [87].

**Figure 24 sensors-19-00055-f024:**
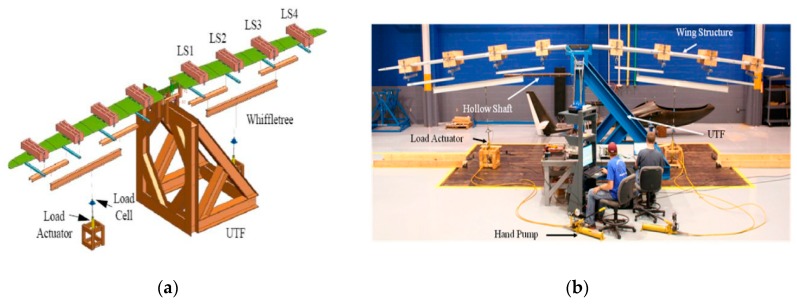
(**a**) Sketch of test for concentrated and distributed loading with the wing loading stations; (**b**) Photo of wing structure under whiffletre loading [90].

**Figure 25 sensors-19-00055-f025:**
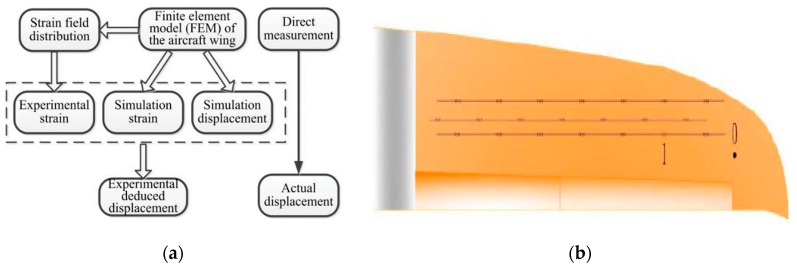
(**a**) Flow chart of deformation measurement; (**b**) Schematic diagram of location of experimental points [95].

**Figure 26 sensors-19-00055-f026:**
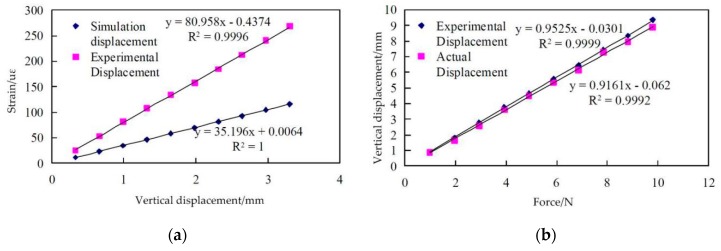
(**a**) Relationship between vertical displacement and strain at Point 1; (**b**) Comparison between actual displacement and experimental displacement at Point 1 [95].

**Figure 27 sensors-19-00055-f027:**
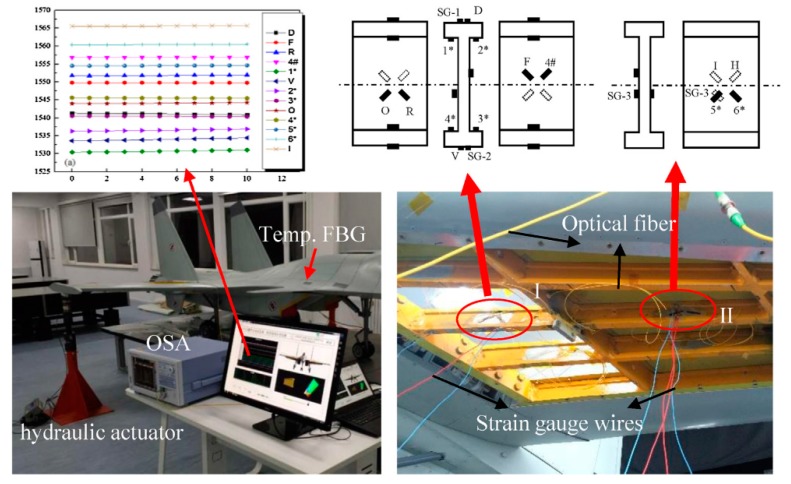
Experimental equipment and sensor installation locations [96].

**Table 1 sensors-19-00055-t001:** Analysis of influencing factors.

Influencing factors	Relation
Adhesive layer	Inverse ratio
Length of sensor grating area	Direct ratio
Elastic modulus of adhesive layer	Direct ratio

**Table 2 sensors-19-00055-t002:** Analysis of the characteristics of three measuring methods.

Method category		Advantages	Disadvantages	Whether it Suitable for Full field Measurement	Whether it Suitable for High Altitude Flight
**Measuring deformation with strain gages**		-Long life -Good frequency response characteristics	-Long time installation -Large volume -Additional load - High cost	No	No
**Measuring deformation with laser**		-High precision -Good real-time property	-Complex device -Poor stability, -Partial area measurement	No	No
**Visual deformation measurement**	**VMD** **measurement system**	-Easy to implement, -High precision	-Insufficient measuring points	No	No
	**PMI** **measurement system**	-High precision, -Full field measurement	-Susceptible to light -Limited to specific ground deformation measurements	Yes	No
	**IPCT** **measurement system**	-High precision, -Flexible grating division -Full field measurement	-Influenced by noise and complex weather conditions	Yes	No

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
