# Peer review of "Fiber Bragg Gratings Sensors for Aircraft Wing Shape Measurement: Recent Applications and Technical Analysis"

_sensors, 2018, doi:10.3390/s19010055_

Reviewer 1 Report

      The authors present a review of aircraft wing deformation using fiber optic sensors (FOSs). Overall, the use and application of FOSs for aicraft monitoring, in particular wings, is an interesting research topic due to the potential benefits and advantages of these sensors over more traditional techniques. However, some areas of the paper need clarification and improvement. The authors are invited to consider the following points, make suitable modifications and resubmit the paper.  

      - The authors target "wing deformation" as an application, and in particular aircraft wings; this should already be clear in the title ("aircraft wing deformation"); also, "deformation" is quite generic: it may refer to load or vibration measurement and may be useful for shape or defect detection, for on-ground or in-flight tests; it is suggested to be more specific on the application cases and related specifications that the authors wish to discuss in their paper

      - there exist different types of fiber optic sensors, but the focus of the paper appears to be specifically on Fiber Bragg Gratings sensors (FBGs): this should also be made clear in the title and abstract

      - "Introduction" section: although there is certainly interesting work ongoing in the US on the subject of the paper, the topic is of general interest and there is a great amount of research in Europe or Asia as well; it is suggested to re-write this section using some of the mentioned cases (US aircrafts) just as possible examples

      - Table 1, page 4: this table is rather unclear and incomplete: there are many more fiber optic sensors types (for example long period or chirped FBGs) and application areas as well as optical characteristics to be considered - it is suggested to remove the table and be more clear on the type of FBG targeted in the paper together with their main features and advantages

      -  Sections 2 and 3, particularly on multiplexing technologies and strain transfer of adhesively-bonded FBGs, appear to be weakly linked to the main focus of the paper, i.e. wing deformation monitoring; the authors should explain when and where multiplexing can be useful in relation to the specific application discussed in the paper (please refer to the first point cited above) and add some considerations on the technologies available for multiplexing (there has been much reaserch going on to optimise interrogators depending on the number of sensors needed and the frequency range of interest); the sections on strain transfer is also quite generic and should be better linked to the requirements of the target applications - for example, in the case of composite structures the sensors can also be embedded in the material; however, there is no mention on that in the paper, whilst composite materials are increasingly used for aircraft structures

      - the term "gate" is used extensively in the paper; does it refer to the grating area?

      - Section 4 on other measurement methods: not all the cited methods are suitable for in-flight measurements; in this case too the authors should be more clear regarding the objective of their discussion: are they considering sensing technologies useful for on-ground tests (like wind tunnel tests, for example) or for real-time, in-flight monitoring of wings?

      - Section 5: the work cited refer to different application cases; each case should be discussed in more depth considering the test purpose, the interrogator and type/number of sensors used (this information is not always discussed), whether they were bonded and how, etc.

       - "Conclusions" section: the remarks appear to be weakly related to the overview reported in the previous sections mainly due to lack of information (see point above); it is suggested to re-write this section based on a more critical description of the work cited in bibliography with a more clear identification of the monitoring purposes.

Author Response

Dear Reviewer:

Thank you very much for your comments.We would like to submit the revised manuscript entitled “Fiber Bragg Gratings Sensors for Aircraft Wing Shape Measurement: Recent Applications and Technical Analysis” for consideration by “Sensors”. We would like to thank your thoroughly reviewing on our manuscript and making thoughtful comments and valuable suggestions. We have revised the manuscript to address your comments. Here are the attached Word file.

Please review the manuscript.

Thank you!

Reviewer 2 Report

      While in principle the subject of the article merits a review paper, this article fails to deliver a comprehensive overview of the field. While the presented paper has some merits on balance I do not consider it of the standard appropriate for a good journal and it would need a total rewrite beyond what is considered major revisions to be of good enough quality.

      This is because some major flaws, such as some important content missing, for example in Sec. 2.3. Also the article is often not well formatted, has low-quality figures, English language problems, many sentences that don't make sense and sometimes references are not in the right places or missing. Also the article carries too many qualitative judgments, for example “high sensitivity”, instead of quoting at least some quantitative numbers from the reference in question that lets the reader compare the performance of the approaches, which is, in my opinion, one of the main tasks that a review paper has to fulfil to be useful.

      Typical examples of the mentioned flaws (this is not a correction list as there are more flaws in the paper that are not listed here):

        - P4: Table 1 has no references, the year-numbers and the application areas are strangely specific, where I would consider it very unlikely that for example 1550nm FBGs have only been used since 2008, certainly such a claim cannot be made without references.

      - P4: The numbered list on page 4 contained several points that do not make sense, both in terms of English language or scientific content,, or the list is not properly formatted. Questionable phrases include “No fever or radiation”, “Passive sensing and it can…”, what do the authors mean by “Parametric induction”, I have never heard of this term. Also, the list fails to mention any problem areas of FBGs, such as temperature/strain cross-sensitivity.

      - Sec. 2.1 has a basic discussion of optical fibers and FBGs but forgets to mention the difference between single-mode and multi-mode fibers as well as making it clear that FBGs are generally only possible in single-mode fiber.

      - P7L179: What does a sentence like “…the same measuring point with different positions of large aircraft wing structure, greatly improving measurement the measurement accuracy and efficiency” mean, it does not make sense in terms of language and contents and uses several unquantified statements.

      - Sec 2.3 This section fails to inform the reader who many FBGs can typically be multiplexed using WDM (<=50). Also they completely fail to mention or explain the other dominant multiplexing technique for FBGs, i.e. OTDR/OFDR.  This is important because these techniques are used in some of the works discussed in later sections, such as Reference [92] where 780 FBGs were multiplexed were interrogated in this way .Also this section fails to discuss the complexity, robustness and price point of the various interrogation approaches.

      - Fig7: The spectra are not at all how a typical FBG spectrum would look like (Gaussian)

      - Sec 2.3.2 The setup in the figure also relies on WDM and has does not describe any spatial multiplexing since all reflected FBG spectra to the same detector and rave the same range and thus cannot be distinguished other than on their wavelength.

      - Sec. 4.2 again no quantities figures for the mentioned laser techniques that let the reader have an idea of what performance can be achieved. This is also true for Table 3.

      - The whole discussion in Section 5 again does not provide any attempt a comparison of the merits of the reviewed techniques, it just lists and summarised the papers without critical discussion or comparison.

Author Response

Dear Reviewer:

Thank you very much for your comments.We would like to submit the revised manuscript entitled “Fiber Bragg Gratings Sensors for Aircraft Wing Shape Measurement: Recent Applications and Technical Analysis” for consideration by “Sensors”. We would like to thank your thoroughly reviewing on our manuscript and making thoughtful comments and valuable suggestions. We have revised the manuscript to address your comments. Here is the attached Word file.

Please review the manuscript.

Thank you!

Round  2

Reviewer 1 Report

      The authors addressed all the points raised in the review report appropriately. The quality of the manuscript has improved making it suitable for publication.
